# Severe Parainfluenza Viral Infection—A Retrospective Study of Adult Intensive Care Patients

**DOI:** 10.3390/jcm12227106

**Published:** 2023-11-15

**Authors:** Adam Watson, Ryan Beecham, Michael P. W. Grocott, Kordo Saeed, Ahilanandan Dushianthan

**Affiliations:** 1General Intensive Care Unit, Southampton General Hospital, Southampton SO16 6YD, UK; a.watson@soton.ac.uk (A.W.); ryan.beecham@uhs.nhs.uk (R.B.); kordo.saeed@uhs.nhs.uk (K.S.); a.dushianthan@soton.ac.uk (A.D.); 2Perioperative and Critical Care Theme, NIHR Southampton Biomedical Research Centre, University Hospital Southampton NHS Foundation Trust, Southampton SO16 6YD, UK; 3Faculty of Medicine, University of Southampton, Southampton SO16 6YD, UK; 4Department of Microbiology, Southampton General Hospital, Southampton SO16 6YD, UK

**Keywords:** parainfluenza, viral, pneumonia, respiratory failure, intensive care

## Abstract

There is little known about parainfluenza virus (PIV) infection in adult intensive care unit (ICU) patients. Here, we aim to describe the characteristics, clinical course and outcomes of PIV infection in adults requiring intensive care. In this retrospective study of consecutive patients admitted to our ICU with confirmed PIV infection over a 7-year period, we report the patient characteristics, laboratory tests and prognostic scores on ICU admission. The main outcomes reported are 30-day mortality and organ support required. We included 50 patients (52% male, mean age 67.6 years). The mean PaO_2_/FiO_2_ and neutrophil/lymphocyte ratios on ICU admission were 198 ± 82 mmHg and 15.7 ± 12.5. Overall, 98% of patients required respiratory support and 24% required cardiovascular support. The median length of ICU stay was 5.9 days (IQR 3.7–9.1) with a 30-day mortality of 40%. In conclusion, PIV infection in adult ICU patients is associated with significant mortality and morbidity. There were significant differences between patients who presented with primary hypoxemic respiratory failure and hypercapnic respiratory failure.

## 1. Introduction

### 1.1. Background

Acute respiratory failure secondary to viral infection is relatively common in the intensive care setting. The prevalence of viral infection among intensive care unit (ICU) patients with pneumonia has been reported as 23–36% [1,2,3]. Whilst influenza, rhinovirus, and, more recently, severe acute respiratory syndrome coronavirus 2 (SARS-CoV-2) are the most frequently detected viral infections in ICU patients, human parainfluenza virus (PIV) is an under-recognised cause of acute respiratory disease among adults [1,2,3].

PIV is a single-stranded ribonucleic acid virus from the Paramyxoviridae family with four recognised serotypes [4,5]. The pathogenesis of PIV infection varies between serotypes, where PIV-1 and PIV-2 are frequently associated with upper respiratory tract infections, whilst pneumonia typically occurs secondary to PIV-3 infection [4,5]. The most common clinical phenotype of PIV infection in adults is a mild upper respiratory tract infection. However, serious lower respiratory tract infections and exacerbations of underlying lung diseases such as chronic obstructive pulmonary disease (COPD) are increasingly recognised [6,7]. Although the symptoms are self-limiting in most cases, risk factors for severe PIV infection in adults are well understood to include older age, cardiac or respiratory comorbidities, and immunocompromise [6,8,9]. The disease burden of PIV is significant, with a reported 3% rate of PIV infection among hospitalised adults with pneumonia [10]. Furthermore, PIV infection may account for up to 12–23.5% of all ICU admissions with viral pneumonia [2,3,11].

However, little is known about the characteristics, organ support requirements and outcomes of ICU patients with severe PIV infection. In the largest study of hospitalised adults with PIV, 23.5% of patients required ICU admission and 18.2% required ventilatory support [6]. In a similar recent study of hospitalised adults with PIV infection, 11.6% of patients required ventilatory support and 1.5% needed cardiovascular support [12]. However, studies to date have focused primarily on hospitalised adults, and PIV infection in ICU patients has yet to be fully described.

### 1.2. Aim and Objectives 

We aim to describe the characteristics and outcomes of PIV infection in adults requiring ICU care. Our main objective is to report the clinical course, organ support requirements, morbidity, and mortality of PIV infection within this setting. Our secondary objective is to investigate specific factors that may predict PIV infection-associated mortality and morbidity.

## 2. Materials and Methods

### 2.1. Study Design and Setting

In this retrospective study, we included consecutive adults with PIV infection requiring ICU admission in a large tertiary hospital in the south of England. The study data were collected for the period between 1 January 2016 and 30 June 2023. We excluded patients with incomplete or unretrievable patient records. This study was sponsored by the University Hospital Southampton NHS Foundation Trust (RHM CRI 0370) and ethical approval was obtained from the NHS Health Research Authority (IRAS 232922). Consent was waived due to the retrospective observational nature of this study. This study is compliant with local ethical standards and no identifiable patient data are presented here. This manuscript complies with STROBE guidelines [13].

### 2.2. Data Collection

PIV-positive patients were identified retrospectively from laboratory polymerase chain reaction (PCR). We used an in-house PCR assay that targets the nucleocapsid protein for parainfluenza 1 and haemagglutinin-neuraminidase gene for parainfluenza 2 and 3. The primers were obtained from Integrated DNA Technologies (IDT), USA and Applied Biosystems (ThermoFisher Scientific, Waltham, MA, USA). The assay is designed as a triplex with each of the targets represented on a different fluorophore allowing independent detection of each target. All patients testing positive for PIV on RT-PCR were included. However, our laboratory does not routinely test for the PIV-4 serotype. Anonymised patient data were then retrieved from electronic patient records (MetaVision CIS, iMDsoft, Tel Aviv, Israel). This included demographic information, comorbidities (described using the Charlson Comorbidity Index) [14] and laboratory values on ICU admission. We categorised comorbidities as cardiovascular (any chronic cardiovascular disease, excluding hypertension), respiratory (any chronic respiratory disease), renal (any chronic kidney disease), neurological (any chronic neurological disease, including dementia and previous cerebrovascular events), solid organ malignancy (current or treated), haematological malignancy (current or treated), and diabetes mellitus. Acute Physiology and Chronic Health Evaluation 2 (APACHE II) and Sequential Organ Failure Assessment (SOFA) scores were calculated for all patients on ICU admission [15,16]. We also collected details of organ support and antimicrobial therapy received. Respiratory support was defined as high-flow nasal oxygen (HFNO), non-invasive ventilation (NIV) as either continuous positive airway pressure (CPAP) or bi-level positive airway pressure (BIPAP), or invasive mechanical ventilation (IMV). The primary outcome reported is 30-day mortality from ICU admission. Our secondary outcome measures are organ support required, incidence of acute kidney injury (AKI), ICU days, hospital days and in-ICU mortality. AKI was defined according to the Kidney Disease Improving Global Outcomes (KIDGO) criteria [17]. 

### 2.3. Statistical Analysis 

Our data are reported using conventional descriptive statistics, with categorical data presented as the number (percentage). We used the Kolmogorov–Smirnov test to assess continuous data for normality, with normally distributed variables presented as the mean ± standard deviation and non-normally distributed variables presented as the median (inter-quartile range). Comparisons were made between survivors and non-survivors at 30 days, and between patients who presented to ICU with hypoxemic respiratory failure and acute hypercapnic respiratory failure. Acute hypercapnic respiratory failure was defined as a PaCO_2_ > 45 mmHg with acidosis, and hypoxemic respiratory failure as PaO_2_/FiO_2_ < 300 mmHg without hypercapnia. Student’s *t* test and the Mann–Whitney U test were used to compare between normally and non-normally distributed data, respectively. We used Fisher’s exact test to compare proportions between groups. Kaplan–Meier curves are used to report 30-day mortality. A *p*-value of <0.05 was taken to be statistically significant.

## 3. Results

### 3.1. Patient Characteristics 

We identified 60 patients from a 7-year period and included 50 in the analysis. There were nine patients excluded due to unretrievable or incomplete electronic patient records. An additional one patient was excluded as a repeat parainfluenza PCR was negative and the contemporaneous clinical opinion was that PIV infection was unlikely (Figure 1). 

The mean age was 67.6 ± 15.5 years with an equal sex distribution (52% male) (Table 1). Mean Charlson Comorbidity Index (CCI) was 4.7 ± 2.3, with 49 patients (98%) reporting one or more significant comorbidity. The majority had respiratory disease (70%), with COPD accounting for 65% of all respiratory comorbidities. Other comorbidities included cardiovascular disease (36%), malignancy (26%), neurological disease (26%) and diabetes mellitus (20%). In addition, nine patients (18%) were on immunosuppressive medications (either systemic chemotherapy or long-term corticosteroids). The average BMI was 27.9 ± 9.9 kg/m^2^ and 48% of patients were current or former smokers. Mean APACHE II and SOFA scores on ICU admission were 16.5 ± 5.1 and 4.2 ± 2.5, whilst the mean PaO_2_/FiO_2_ and neutrophil/lymphocyte (N/L) ratios were 198 ± 82 mmHg and 15.7 ± 12.5, respectively.

### 3.2. Outcome-Mortality 

Among the 50 patients studied, the overall 30-day mortality was 40% (*n* = 20). We noted significant variation in some variables between survivors and non-survivors (Table 1). Overall, the survivors were younger (62.3 vs. 75.3 years, *p* = 0.003) and had lower CCI scores (4.0 vs. 5.8, *p* = 0.007) when compared to non-survivors. Furthermore, cardiovascular disease was less prevalent in survivors compared to non-survivors (23% vs. 55%, *p* = 0.035). However, the prevalence of diabetes mellitus was lower in non-survivors (5% vs. 30%, *p* = 0.037). There were no differences in laboratory tests on ICU admission between survivors and non-survivors (Table 1). We did not identify any differences in severity scores on ICU admission between survivors and non-survivors, although there were non-significant trends towards increased mortality with higher APACHE II scores (*p* = 0.09) and increased N/L ratio (*p* = 0.05). There were 16 patients (32%) who died on ICU and 3 patients (6%) received cardiopulmonary resuscitation (CPR). 

### 3.3. Type of Respiratory Failure

We categorised patients according to type of respiratory failure on ICU admission (Table 2). There were 21 patients (42%) who presented with hypoxemic respiratory failure with a mean P/F ratio, pH and PaCO_2_ of 142 ± 62 mmHg, 7.433 ± 0.080 and 32.3 ± 6.8 mmHg. In contrast, 25 patients (50%) who presented with hypercapnic respiratory failure with a higher mean P/F ratio, lower pH and higher PaCO_2_ of 234 ± 71 mmHg, 7.317 ± 0.106 and 63.0 ± 16.5 mmHg, respectively (*p* < 0.001). There were four patients (8%) who did not initially present with respiratory failure. In comparison to hypercapnic failure, patients with hypoxemic respiratory failure were more immunosuppressed (38% vs. 4%, *p* = 0.0067) and had increased incidence of haematological malignancy (24% vs. 0%, *p* = 0.015) (Table 2). However, patients with hypercapnic respiratory failure had more respiratory comorbidity (96% vs. 43%, *p* < 0.001), particularly COPD. There was no difference in mortality between types of respiratory failure (Figure 2). 

### 3.4. PIV Serotypes

PIV-3 represented 82% of infections, followed by PIV-1 (14%) and PIV-2 (4%). There was a seasonal variation in patients with PIV on our ICU, with a peak around February to May (Figure 3). Furthermore, the distribution of PIV serotypes varied between the type of respiratory failure (Table 2). All patients with hypoxemic respiratory failure tested positive for PIV-3. In contrast, 68% of patients with hypercapnic respiratory failure had PIV-3, whilst 24% had PIV-1 and 8% had PIV-2. These differences were significant for PIV-1 (*p* = 0.025) and PIV-3 (*p* = 0.005). However, there were no differences in PIV serotypes between survivors and non-survivors. 

### 3.5. Clinical Course

We describe the clinical course of PIV infection within our ICU as median days from admission to key events (Table 3). The overall median time from hospital admission to ICU admission was 0.4 days (IQR 0.0–9.3), with no difference between survivors and non-survivors (*p* = 0.47). There were 27 patients (52%) admitted directly from the Emergency Department (ED) or Acute Medical Unit, and the proportion of patients admitted from ED was higher for survivors compared to non-survivors (47% vs. 15%, *p* < 0.01). Furthermore, median time to ICU admission for hypoxemic respiratory failure was longer than for hypercapnic respiratory failure (3.3 days vs. 0.2 days, *p* = 0.034). However, within our ICU, there were no differences in the clinical course between types of respiratory failure (Table 3). The median time to death and ICU discharge were 4.8 days (IQR 1.9–11.5) and 5.8 days (IQR 3.7–9.1), respectively. 

### 3.6. Respiratory Support

Overall, 49 patients (98%) required respiratory support (Table 4). There were 24 patients (48%) who only received one type of respiratory support, with the remainder of patients receiving two (*n* = 21, 42%) or three (*n* = 4, 8%) types of respiratory support. Of patients with hypoxemic respiratory failure, the first types of respiratory support were NIV (*n* = 9, 43%), HFNO (*n* = 9, 43%) and IMV (*n* = 3, 14%). In contrast, patients with hypercapnic respiratory failure, the first types of respiratory support were NIV (*n* = 21, 84%), HFNO (*n* = 2, 8%) and IMV (*n* = 2, 8%). The use of HFNO use at any point was more frequent in patients with hypoxemic respiratory failure compared to hypercapnic respiratory failure (82% vs. 28%, *p* < 0.001). However, there were no other differences in organ support required or number of organs supported between types of respiratory failure (Table 4).

There were 41 patients (82%) who received NIV at any point, with a median time to NIV from ICU admission of 2 h (IQR 1–6). Of these patients, seven (14%) primarily received CPAP with a mean starting pressure of 7 ± 2 cmH_2_O and an FiO_2_ 56 ± 24%. A further 34 patients (68%) received Bilevel support with mean starting settings of IPAP 15 ± 5 cmH_2_O, EPAP 6 ± 2 cmH_2_O and FiO_2_ 37 ± 15%. Of the 30 patients (60%) who received NIV as their initial respiratory support, 9 patients (18%) did not tolerate it and were either intubated (*n* = 3, 6%) or died (*n* = 6, 12%). The 30-day mortality of patients who received NIV as their initial respiratory support was 40% (*n* = 20). 

There were 11 patients (22%) who received IMV, with a median time to IMV start of 1 h (IQR 1-1). Mean SOFA score on ICU admission for patients who received IMV was greater compared to patients who did not receive IMV (6.6 vs. 3.5, *p* < 0.001). The mean IMV starting settings were a FiO_2_ of 50 ± 9.7%, Positive Inspiratory Pressure of 26 ± 5 cmH_2_O and Positive End Expiratory Pressure of 8 ± 2 cmH_2_O, with a mean tidal volume of 7.2 ± 1.5 mL/kg of ideal body weight. Of patients who received IMV, the mean lowest P/F ratio was 131 ± 39 mmHg and three (27%) died within 30 days.

### 3.7. Other Organ Support 

There were 12 patients (24%) who received cardiovascular (CV) support in the forms of either norepinephrine or vasopressin. Mean admission SOFA for patients requiring CV support was higher than those not requiring CV support (7.0 vs. 3.0, *p* < 0.001). The median duration of CV support was 5 days (IQR 2–7.3). Overall, there were 14 patients (28%) treated for AF with RVR, 12 patients (24%) treated for decompensated heart failure, and 9 patients (18%) with evidence of myocardial ischemia. We measured Troponin I for 10 patients (20%), with a median peak of 556 ng/mL (IQR 196–871).

In total, 21 patients (42%) developed an acute kidney injury (AKI) with a median peak creatinine of 104 μmol/L (IQR 74–145). However, only three patients (6%) required renal replacement therapy (RRT) for a median time of 8.0 days (IQR 5.5–11.0). Mean age and APACHE II score were higher in patients with an AKI compared to those without (72.6 vs. 63.9 years, *p* = 0.034; 18.2 vs. 15.2, *p* = 0.037). The mean fluid balance on day 4 and on ICU discharge were +911 ± 4183 mL and +242 ± 5386 mL, respectively, with no difference between patients with or without an AKI. However, patients with an AKI were more likely to have received furosemide (71% vs. 17%, *p* < 0.001). Overall, although statistically not significant, if a patient developed an AKI, the odds ratio for death at 30 days was 2.36 (95% CI 0.70–7.93, *p* = 0.17). 

### 3.8. Infection Markers and Microbiology 

The mean peak WCC and CRP were 18.6 ± 10.8 × 10^9^/L and 177 ± 142 mg/L, with no difference between survivors and non-survivors (Table 1). Mean peak temperature was 37.6 ± 1.0 °C, with 11 patients (22%) recording a temperature over 38 °C. Overall, 94% of patients were treated with antibiotics. There were six patients (12%) who received oseltamivir, one of whom had influenza co-infection, and three patients (6%) treated with antifungals. There were 29 patients (58%) treated with steroids, which had no association with 30-day mortality (*p* = 0.56). Overall, 12 patients (24%) had proven respiratory co-infection. The viral co-infections were rhinovirus (*n* = 3), metapneumovirus (*n* = 1), respiratory syncytial virus (*n* = 1), influenza A (*n* = 1), and SARS-CoV-2 (*n* = 1). The bacterial or fungal co-infections were *pseudomonas aeruginosa* (*n* = 2), *klebsiella pneumoniae* (*n* = 1), *Escherichia coli* (*n* =1), *citrobacter koseri* (*n* = 1), and *aspergillus* (*n* = 1). There was no association between respiratory co-infection and 30-day mortality (*p* = 0.89) or co-infection and requirement for cardiovascular support (*p* = 0.51) or renal replacement therapy (*p* = 0.77).

## 4. Discussion

In this study, we report the characteristics, organ support requirements and outcomes of severe PIV infection in our ICU over 7 years. The common organ dysfunction was acute respiratory failure. Although there were no significant differences in mortality, there were substantial clinical variations between patients admitted with hypoxemic and hypercapnic respiratory failure. Most patients needed invasive or non-invasive respiratory support, with an overall mortality of 40%. As far as we know, this is the first study to describe in detail the characteristics, clinical course, and outcomes of PIV infection in adult ICU patients. 

Viral pneumonia is a common ICU presentation, accounting for approximately a third of all patients with pneumonia [1,2]. The recent COVID-19 pandemic exposed the limitations of healthcare services to provide appropriate organ support for emerging viral infections [18]. Although some guidelines exist, studies addressing the specific management of patients with severe viral pneumonia are lacking. Furthermore, there are substantial variations in clinical presentations, treatment strategies, and outcomes between patients with different viral aetiologies [19]. As authors to date have focused primarily on hospitalised adults with PIV infection, studies focusing on patients requiring ICU admission and organ support are needed to clearly define this population for the future. 

As expected, patients with severe PIV infection are elderly and have significant comorbidities, most prominently respiratory disease (70%), malignancy (26%) and immunosuppression (18%). These factors have previously been associated with severe PIV infection [6,8]. Our data also suggest that adults with severe PIV infection who present to the ICU can be categorised by type of respiratory failure. In patients with hypoxaemic respiratory failure, we found that the prevalence of haematological malignancy, immunosuppressive medication and diabetes mellitus was higher than for hypercapnic respiratory failure. In contrast, the prevalence of chronic respiratory disease, particularly COPD, was higher in patients with hypercapnic respiratory failure. PIV-3 was the most common serotype identified in our ICU, in keeping with previous epidemiological studies [3]. We also noted that PIV-3 exclusively caused hypoxaemic respiratory failure, whereas the prevalence of PIV-1 was greater for hypercapnic respiratory failure. The pathogenesis of PIV serotypes may explain these differences, as PIV-1 and PIV-2 are frequently associated with tracheobronchitis, whilst PIV-3 is more likely to cause bronchiolitis or pneumonia [4]. Furthermore, the prevalence of proven respiratory co-infection was 24% in this cohort, although there was no association with increased mortality, which has previously been reported [6].

In this study, 98% of patients required respiratory support. We noted that 82% of patients received NIV, often as their first type of ventilatory support, regardless of the type of respiratory failure. Although using NIV for pneumonia and acute hypoxemic respiratory failure remains controversial, it is increasingly recognised as a non-invasive respiratory support tool to improve oxygenation in pneumonia and ARDS [20,21,22]. Our results suggest that, on average, patients with severe PIV infection require NIV for 3.0 days and IMV for 8.4 days. However, the average time from hospital presentation to ICU admission varied between hypoxaemic at 3.3 days and hypercapnic at 0.2 days. This may reflect both a slower progression of viral pneumonia in patients with hypoxaemic respiratory failure and the relative ease of identifying the need for ICU care, such as NIV, in patients with hypercapnic respiratory failure. 

The 30-day mortality in this cohort was 40%, which is higher than previously reported for PIV infection requiring ICU care [3]. This may be partially attributable to differences between healthcare systems in their approach to ICU care or differences in patient characteristics and presentation but highlights the need for future studies to investigate this population further. The most significant risk factors for mortality appear to be age and co-morbid status, particularly cardiovascular disease, which was more prevalent in non-survivors. However, in contrast to previous studies [8,9], there was no association between malignancy or immunosuppressive medication and mortality. We also report significant morbidity, for example, AKI and myocardial ischemia, which had a prevalence of 42% and 18%, respectively.

There was a trend towards increased mortality with higher N/L ratio on ICU admission. N/L ratio is an emergency biomarker reflecting the balance between inflammation and adaptive immunity [23]. The prognostic value of increased N/L ratio has been well studied for COVID-19 pneumonia [24], with a N/L ratio of >6.5 strongly associated with disease severity and mortality [25]. In this cohort, the mean N/L ratio on ICU admission was 15.7, which may partially explain the high mortality reported here. Of the other prognostic scores reported here, APACHE II was significantly higher in patients who developed an AKI and trended to higher in non-survivors. Furthermore, SOFA scores were greater in patient requiring IMV and CV support, which may suggest that our ICU was more likely to offer these therapies to patient with potentially reversible sepsis, as opposed to worsening and likely terminal respiratory failure. Our findings suggest that N/L ratio should be investigated further as a prognostic marker in patients with severe viral pneumonia.

Our study has several limitations. This is a single-centre retrospective study, where despite a long data collection period, our sample size remained limited. Furthermore, there was significant heterogeneity between patients with variations in severity and type of respiratory failure. As our study did not include hospitalised patients with PIV infection, we are unable to quantity the overall prevalence of severe PIV infection in our hospital. Our results should also be interpreted in the context of how ICU care is utilised in the United Kingdom. We routinely discuss limitations of care with patients that incorporates their wishes, pre-morbid functional status and frailty, and the ability to recover from critical illness and invasive organ support. Therefore, some patients with severe PIV infection may have not admitted to ICU to receive specific interventions (e.g., IMV), even if their disease severity alone might have warranted this. It was not feasible to adjust for this in our analysis, but we have highlighted when appropriate how our practice may have influenced results. We were also unable to accurately report the pre-morbid functional status of our cohort, although this would have influenced decisions regarding ICU admission and management. Moreover, we were unable to present the incidence of PIV-4 as our laboratory does not routinely test for PIV-4 serotype. We are also not able to report PCR cycle threshold (CT-PCR) data or its potential association with mortality due to inconsistent availability of data over the 7-year period. However, despite these limitations, we are the first to report specifically on the characteristics and outcomes of PIV patients admitted to ICU for organ support. Our results affirm the importance of age and comorbid status as risk factors for morbidity in severe PIV infection. They also suggest that severe PIV infection presents either as acute hypoxemic or hypercapnic respiratory failure, with separate risk factors for each. 

## 5. Conclusions

We have described PIV infection in an ICU setting in detail for the first time. In this cohort, patients with severe PIV infection were elderly, co-morbid and frequently reported chronic respiratory illness or immunocompromise. While most presented with acute respiratory failure, there were differences between patients presenting with primary hypoxemic and hypercapnic respiratory failure. Nearly all required either non-invasive or invasive respiratory support. The average ICU stay is 5.9 days, with a 30-day mortality of 40%. Other prospective cohort studies are needed to characterise this population in more detail and inform future management strategies.

## Figures and Tables

**Figure 1 jcm-12-07106-f001:**
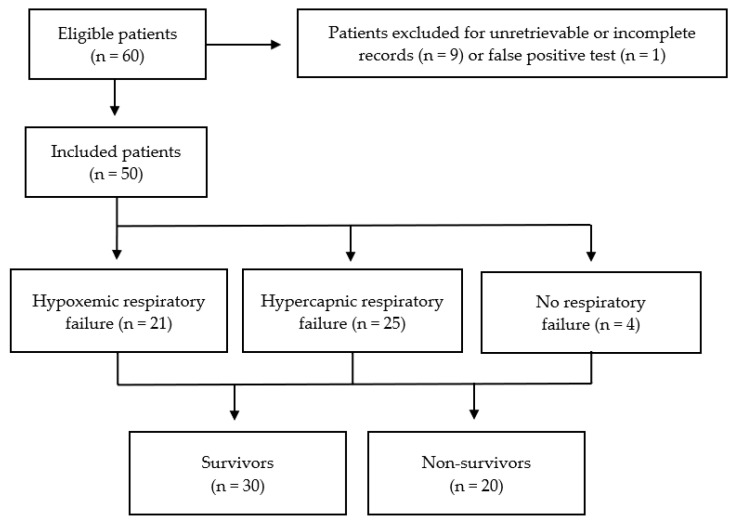
Flow diagram of eligible, included and excluded patients.

**Figure 2 jcm-12-07106-f002:**
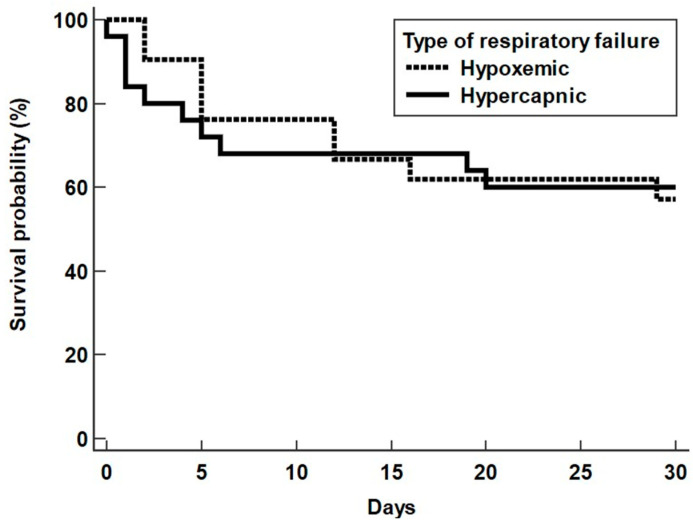
Kaplan–Meier survival curve for types of respiratory failure.

**Figure 3 jcm-12-07106-f003:**
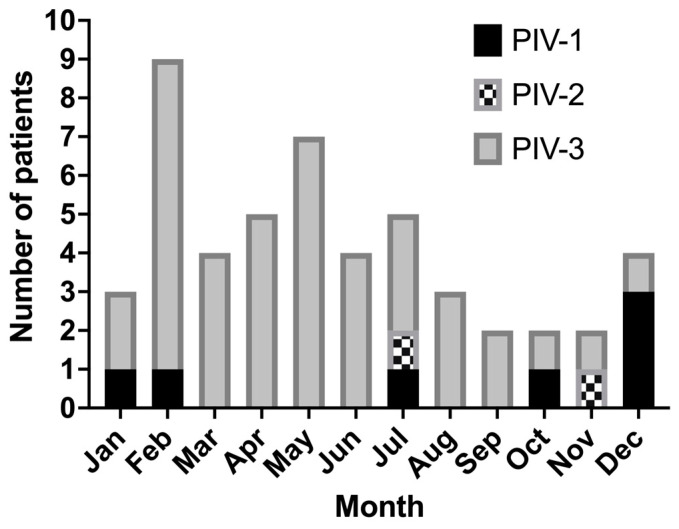
Histogram of included patients by PIV serotypes and month of year.

**Table 1 jcm-12-07106-t001:** Patient characteristics, laboratory tests according to mortality at 30 days post ICU admission.

Characteristics	All (*n* = 50)	Survivors (*n* = 30)	Non-Survivors (*n* = 20)	*p*
Age (years)	67.6 ± 15.5	62.3 ± 16.8	75.3 ± 9.3	* 0.003
Sex, no (%)				
Male	26 (52)	15 (50)	11 (55)	0.78
Female	24 (48)	15 (50)	9 (45)	0.78
Charlson Comorbidity Index	4.7 ± 2.3	4.0 ± 2.1	5.8 ± 2.1	* 0.007
Comorbidities, no (%)				
Any cardiovascular	18 (36)	7 (23)	11 (55)	* 0.035
Any respiratory	35 (70)	21 (70)	14 (70)	1.00
Any renal	4 (8)	2 (7)	2 (10)	1.00
Any neurological	13 (26)	5 (17)	8 (40)	0.10
Solid organ malignancy	8 (16)	3 (10)	5 (25)	0.24
Haematological malignancy	5 (10)	3 (10)	2 (10)	1.00
Diabetes mellitus	10 (20)	9 (30)	1 (5)	* 0.037
Laboratory tests on ICU admission				
PaO_2_/FiO_2_ ratio (mmHg)	198 ± 82	209 ± 38	182 ± 85	0.28
PaCO_2_ (mmHg)	51 ± 17	51 ± 15	50 ± 21	0.92
Neutrophil/Lymphocyte ratio	15.7 ± 12.5	12.9 ± 11.8	19.9 ± 12.7	0.05
WCC (109/L)	13.2 ± 7.0	12.3 ± 7.9	14.5 ± 5.5	0.29
CRP (mg/L)	128 ± 108	121 ± 119	137 ± 92	0.60
Bilirubin (μmol/L)	10 (7–17)	8 (7–15)	14 (8–17)	0.46
Urea (mmol/L)	8.9 ± 4.8	8.0 ± 4.9	10.4 ± 4.2	0.08
Creatinine (μmol/L)	72 (53–94)	64 (48–80)	89 (65–105)	0.75
Prognostic scores on ICU admission				
APACHE II	16.5 ± 5.1	15.5 ± 5.9	18.0 ± 3.3	0.09
SOFA	4.2 ± 2.5	4.3 ± 2.8	4.0 ± 2.0	0.64

Abbreviations: white cell count (WCC), C-reactive protein (CRP), Acute Physiology and Chronic Health Evaluation II (APACHE II), and Sequential Organ Failure Assessment (SOFA). Footnotes: data presented as the number (%), mean ± SD, or median (IQR). * *p* < 0.05.

**Table 2 jcm-12-07106-t002:** Patient characteristics, prognostic scores, and laboratory tests according to type of respiratory failure on ICU admission.

Characteristics	Hypoxemic Respiratory Failure (*n* = 21)	Hypercapnic Respiratory Failure (*n* = 25)	*p*
Age (years)	66.7 ± 18.5	68.7 ± 13.8	0.68
Sex, no (%)			
Male	12 (57)	13 (52)	0.77
Female	9 (43)	12 (48)	0.77
Charlson Comorbidity Index	4.9 ± 2.2	4.9 ± 2.4	0.97
Comorbidities, no (%)			
Any cardiovascular	7 (33)	11 (44)	0.55
Any respiratory	9 (43)	24 (96)	* <0.001
Any renal	4 (19)	0 (0)	* 0.037
Any neurological	6 (29)	7 (28)	1
Solid organ malignancy	4 (19)	4 (16)	1
Haematological malignancy	5 (24)	0 (0)	* 0.015
Diabetes mellitus	5 (24)	3 (12)	* 0.044
PIV serotype			
PIV-1	0 (0)	6 (24)	* 0.025
PIV-2	0 (0)	2 (8)	0.493
PIV-3	21 (100)	17 (68)	* 0.005
Laboratory tests on ICU admission			
PaO_2_/FiO_2_ ratio (mmHg)	142 ± 62	234 ± 71	* <0.001
PaCO_2_ (mmHg)	38 ± 8	63 ± 17	* <0.001
Neutrophil/Lymphocyte ratio	14.0 ± 11.1	16.2 ± 11.7	0.52
WCC (10^9^/L)	11.8 ± 5.6	13.2 ± 7.7	0.51
CRP (mg/L)	157 ± 75	106 ± 131	0.12
Bilirubin (μmol/L)	12 (9–23)	8 (6–17)	0.09
Urea (mmol/L)	9.2 ± 6.0	8.6 ± 4.0	0.68
Creatinine (μmol/L)	79 (65–95)	60 (50–89)	0.11
Prognostic scores on ICU admission			
APACHE II	17.0 ± 5.5	16.4 ± 5.0	0.71
SOFA	5.2 ± 2.4	3.4 ± 2.5	* 0.018

Abbreviations: white cell count (WCC), C-reactive protein (CRP), Acute Physiology and Chronic Health Evaluation II (APACHE II), and Sequential Organ Failure Assessment (SOFA). Footnotes: data presented as the number (%), mean ± SD, or median (IQR). * *p* < 0.05.

**Table 3 jcm-12-07106-t003:** Clinical course of PIV infection according to type of respiratory failure on ICU admission.

Event	Days from ICU Admission—Median (IQR)			*p*
	All Patients (*n* = 50)	Hypoxemic Respiratory Failure (*n* = 21)	Hypercapnic Respiratory Failure (*n* = 25)	
First required NIV	0.0 (0.0–0.2)	0.1 (0.0–0.5)	0.0 (0.0–0.0)	0.10
First required CV support	0.0 (0.0–0.4)	0.1 (0.0–1.6)	0.0 (0.0–0.0)	0.09
First required IMW	0.0 (0.0–1.5)	0.2 (0.0–2.8)	0.0 (0.0–0.1)	0.18
First required HFNO	0.3 (0.0–1.6)	0.1 (0.0–0.4)	0.4 (0.1–1.4)	0.85
Parainfluenza positive	0.3 (0.1–1.5)	0.8 (0.1–3.4)	0.2 (0.0–0.4)	0.42
First required RRT	0.3 (0.2–4.7)	0.2 (0.1–0.3)	n/a	n/a
Last required NIV	3.0 (1.3–4.6)	3.1 (2.4–5.2)	1.8 (0.5–4.1)	0.12
Last required HFNO	4.5 (1.7–6.8)	4.8 (2.3–6.0)	1.7 (1.4–4.4)	0.54
Death	4.8 (1.9–11.5)	5.0 (4.8–12.4)	1.9 (1.1–4.8)	0.16
ICU discharge	5.9 (3.7–9.1)	6.5 (4.7–11.2)	5.2 (2.9–6.3)	0.13
Last required IMV	8.4 (4.1–10.1)	9.8 (8.4–10.4)	8.3 (4.8–11.4)	0.36
Hospital discharge	9.7 (5.3–16.5)	9.7 (7.5–24.8)	9.1 (4.2–13.5)	0.91

Abbreviations: high-flow nasal oxygen (HFNO), non-invasive ventilation (NIV), cardiovascular (CV), invasive mechanical ventilation (IMV), renal replacement therapy (RRT). n/a, not applicable as there were no patients required renal replacement therapy. Footnotes: four patients did not initially present with respiratory failure, so were excluded from sub-groups.

**Table 4 jcm-12-07106-t004:** Organ support requirement and mortality according to type of respiratory failure on ICU admission.

Variable	All Patients (*n* = 50)	Hypoxemic Respiratory Failure (*n* = 21)	Hypercapnic Respiratory Failure (*n* = 25)	*p*
Organ support required—no (%)				
HFNO	26 (52)	17 (82)	7 (28)	* <0.001
NIV	41 (82)	18 (86)	21 (84)	1.0
IMV	11 (22)	5 (24)	3 (12)	0.44
Any respiratory support	49 (98)	21 (100)	24 (96)	1.0
Any cardiovascular support	12 (24)	6 (29)	3 (12)	0.26
Any RRT	3 (6)	2 (10)	0 (0)	0.202
Total organs supported—no (%)				
0	1 (2)	0 (0)	1 (4)	1.0
1	38 (76)	15 (71)	21 (84)	0.47
2	8 (16)	4 (19)	3 (12)	0.69
3	3 (6)	2 (10)	0 (0)	0.20
Mortality—no (%)				
In ICU	16 (32)	8 (38)	7 (28)	0.54
30 days	20 (40)	9 (43)	10 (40)	0.77

Abbreviations: high-flow nasal oxygen (HFNO), non-invasive ventilation (NIV), invasive mechanical ventilation (IMV), and renal replacement therapy (RRT). Footnotes: four patients did not initially present with respiratory failure, so were excluded from sub-groups. * *p* < 0.05.

## Data Availability

Data available in a publicly accessible repository. The data presented in this study are openly available in FigShare at https://doi.org/10.6084/m9.figshare.24559285.v1.

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
