# Peer review of "Severe Parainfluenza Viral Infection—A Retrospective Study of Adult Intensive Care Patients"

_jcm, 2023, doi:10.3390/jcm12227106_

Round 1

Reviewer 1 Report

Comments and Suggestions for Authors

This manuscript has good scientific merit and would appeal to a wide range of readers, including laboratorians (provided the following comments can be addressed).

Line 77: state the molecular target of the RT-PCR. For instance, was the targeted gene coding for the nucleocapsid (N) or the hemagglutinin–neuraminidase surface glycoprotein (HN) or something else?

Line 79: HPIV is an abbreviation which has not been spelt out prior to its first appearance in the text.

Figure 2: consider using a solid and a dotted line to represent the resp. failure types instead of two colours which would appear similar when the article is printed using a B&W printer.

Figure 3: consider using colours which would contrast each other significantly (such as grey, black and white) to accommodate readers who print the article using a B&W printer.

Line 263-4: bacterial and fungal names should be in italics. All letters of each name should be in lowercase, except for the first letter of the genus (in uppercase).

It would be interesting to present the PCR Ct (cycle threshold) values as well, because lower Ct values generally correlate with poorer prognosis/more severe disease.

The authors mentioned that there was no association between respiratory co-infection and 30-day mortality. What about co-infection and organ support requirement?

The discussion put forth by the authors was good. They attempted to explain why PIV-3 and PIV-1 caused hypoxaemic respiratory failure and hypercapnic respiratory failure, respectively.

Comments on the Quality of English Language

The manuscript was well-written and easy to comprehend. 

Author Response

Dear Editor and Reviewers,

Thank you so much for your valuable comments and feedback on our manuscript. We are delighted that you found our results interesting and appealing. Please see below our response to the specific points raised.

Reviewer 1

This manuscript has good scientific merit and would appeal to a wide range of readers, including laboratorians (provided the following comments can be addressed).

  1. Line 77: state the molecular target of the RT-PCR. For instance, was the targeted gene coding for the nucleocapsid (N) or the hemagglutinin–neuraminidase surface glycoprotein (HN) or something else?

Response: We have added this detail to our methods section (lines 77-80).

  1. Line 79: HPIV is an abbreviation which has not been spelt out prior to its first appearance in the text.

Response: We have corrected this.

  1. Figure 2: consider using a solid and a dotted line to represent the resp. failure types instead of two colours which would appear similar when the article is printed using a B&W printer.

Response: We have modified the figure as suggested.

  1. Figure 3: consider using colours which would contrast each other significantly (such as grey, black and white) to accommodate readers who print the article using a B&W printer.

Response: We have modified the figure as suggested.

  1. Line 263-4: bacterial and fungal names should be in italics. All letters of each name should be in lowercase, except for the first letter of the genus (in uppercase).

Response: We have updated these.

  1. It would be interesting to present the PCR Ct (cycle threshold) values as well, because lower Ct values generally correlate with poorer prognosis/more severe disease.

Response: We are unfortunately unable to report PCR Ct data as these records are not fully accessible due to the passage of time. However, we have added this to our limitations and agree it would be an interesting area for future research (380-382).

  1. The authors mentioned that there was no association between respiratory co-infection and 30-day mortality. What about co-infection and organ support requirement?

Response: We have added to our results that there was no association between co-infection and requirement for cardiovascular support or renal replacement therapy (lines 281,282).  

  1. The discussion put forth by the authors was good. They attempted to explain why PIV-3 and PIV-1 caused hypoxaemic respiratory failure and hypercapnic respiratory failure, respectively.

Response: Thank you.

Reviewer 2 Report

Comments and Suggestions for Authors

The manuscript "Severe parainfluenza viral infection - A retrospective study of adult intensive care patients" by Adam Watson, Ryan Beecham, Michael PW Grocott, Kordo Saeed and Ahilanandan Dushian, submitted for review, presents the important issue of PIV infection in patients in intensive care units. Data collected over a seven-year period from 50 patients were analyzed. Authors report the characteristics, organ support requirements and outcomes of severe PIV infection. In my opinion, the presented issue is interesting for readers, the research provides information about the course of infection and prognosis for PIV infected patients hospitalized in the intensive care unit. The authors also discuss the role of the ICU in the treatment of this perhaps underestimated infection. However, the manuscript has some flaws that should be corrected.

1. There is no description of the PIV detection methodology in the materials and methods section, e.g. primers used, target regions, if it has been published, there should be references

2. Table 1 - is first referenced only in the next paragraph, although the data is discussed earlier. Shouldn't there be a reference to table 1 earlier?

3 - lines: 148-151 please explain the number of patients, is it correct with 50 subjects and 40% mortality?

4. -line 79 HPIV instead of PIV and typo in line 361 unable

5. References should be formatted correctly (e.g.: 1. Loubet, P.; Voiriot, G.; Houhou-Fidouh, N.; Neuville, M.; Bouadma, L.; Lescure, F.-X.; Descamps, D.; Timsit, J.-F.; Yazdanpanah, Y.; Visseaux, B. Impact of Respiratory Viruses in Hospital-Acquired Pneumonia in the Intensive Care Unit: A Single-Center Retrospective Study. Journal of Clinical Virology 2017, 91, 52–57, doi:10.1016/j.jcv.2017.04.001.)

Author Response

Dear Editor and Reviewers,

Thank you so much for your valuable comments and feedback on our manuscript. We are delighted that you found our results interesting and appealing. Please see below our response to the specific points raised.

Reviewer 2

The manuscript "Severe parainfluenza viral infection - A retrospective study of adult intensive care patients" by Adam Watson, Ryan Beecham, Michael PW Grocott, Kordo Saeed and Ahilanandan Dushian, submitted for review, presents the important issue of PIV infection in patients in intensive care units. Data collected over a seven-year period from 50 patients were analyzed. Authors report the characteristics, organ support requirements and outcomes of severe PIV infection. In my opinion, the presented issue is interesting for readers, the research provides information about the course of infection and prognosis for PIV infected patients hospitalized in the intensive care unit. The authors also discuss the role of the ICU in the treatment of this perhaps underestimated infection. However, the manuscript has some flaws that should be corrected.

  1. There is no description of the PIV detection methodology in the materials and methods section, e.g., primers used, target regions, if it has been published, there should be references.

Response: We have now added additional details on the methods section (lines 77-80).

  1. Table 1 - is first referenced only in the next paragraph, although the data is discussed earlier. Shouldn't there be a reference to table 1 earlier?

Response: We have added a reference to Table 1 earlier when the data is first reported.

  1. lines: 148-151 please explain the number of patients, is it correct with 50 subjects and 40% mortality?

Response: We have now modified this paragraph clarifying the number of total patients. We have also removed the last sentence to avoid any confusion.

  1. line 79 HPIV instead of PIV and typo in line 361 unable

Response: We have corrected these.

  1. References should be formatted correctly (e.g.: 1. Loubet, P.; Voiriot, G.; Houhou-Fidouh, N.; Neuville, M.; Bouadma, L.; Lescure, F.-X.; Descamps, D.; Timsit, J.-F.; Yazdanpanah, Y.; Visseaux, B. Impact of Respiratory Viruses in Hospital-Acquired Pneumonia in the Intensive Care Unit: A Single-Center Retrospective Study. Journal of Clinical Virology 2017, 91, 52–57, doi:10.1016/j.jcv.2017.04.001.

Response: We have updated the reference style.